# Impact of Daily Bedside Echocardiographic Assessment on Readmissions in Acute Heart Failure: A Randomized Clinical Trial

**DOI:** 10.3390/jcm11072047

**Published:** 2022-04-06

**Authors:** Jean-Etienne Ricci, Sylvain Aguilhon, Bob-Valéry Occean, Camille Soullier, Kamila Solecki, Christelle Robert, Fabien Huet, Luc Cornillet, Laurent Schmutz, Thierry Chevallier, Mariama Akodad, Florence Leclercq, Guillaume Cayla, Benoît Lattuca, François Roubille

**Affiliations:** 1Department of Cardiology, CHU Nîmes, University of Montpellier, CEDEX 9, 30029 Nîmes, France; s-aguilhon@chu-montpellier.fr (S.A.); camille.soullier@chu-nimes.fr (C.S.); christelle.robert@chu-nimes.fr (C.R.); luc.cornillet@chu-nimes.fr (L.C.); laurent.schmutz@chu-nimes.fr (L.S.); guillaume.cayla@chu-nimes.fr (G.C.); benoit.lattuca@chu-nimes.fr (B.L.); 2IMAGINE UR UM 103, Department Cardiology, Nîmes University Hospital, University of Montpellier, 30029 Nîmes, France; 3Department of Cardiology, Montpellier University Hospital, CEDEX 5, 34295 Montpellier, France; kamila4873@hotmail.com (K.S.); f-huet@chu-montpellier.fr (F.H.); akodadmyriam@gmail.com (M.A.); f-leclercq@chu-montpellier.fr (F.L.); f-roubille@chu-montpellier.fr (F.R.); 4Department of Biostatistics, Epidemiology, Public Health and Innovation in Methodology (BESPIM), CHU Nîmes, University of Montpellier, CEDEX 9, 30029 Nîmes, France; bob.occean@chu-nimes.fr (B.-V.O.); thierry.chevallier@chu-nimes.fr (T.C.); 5Physiologie et Médecine Expérimentale du Cœur et des Muscles, INSERM U1046, CNRS UMR 9214, University of Montpellier, CEDEX 5, 34295 Montpellier, France

**Keywords:** acute heart failure, bedside echocardiography, cardiac filling pressure, congestion, readmissions

## Abstract

Acute heart failure (AHF) management is challenging, with high morbidity and readmission rates. There is little evidence of the benefit of HF monitoring during hospitalization. The aim of the study was to assess whether daily bedside echocardiographic monitoring (JetEcho) improved outcomes in AHF. In this prospective, open, two parallel-arm study (clinicaltrials.gov: NCT02892227), participants from two university hospitals were randomized to either standard of care (SC) or daily treatment adjustment including diuretics guided by JetEcho evaluating left ventricular filling pressure and volemia. The primary outcome was 30-day readmission rate. Key secondary outcomes were six-month cumulative incidence death, worsening HF during hospitalization and increasing of myocardial and renal biomarkers. From 250 included patients, 115 were finally analyzed in JetEcho group and 112 in SC group. Twenty-two (19%) patients were readmitted within 30 days in JetEcho group and 17 (15%) in SC group (relative risk [RR] 1.26; 95% confidence interval [CI], 0.70–2.24; *p* = 0.4). Worsening HF occurred in 17 (14%) patients in the JetEcho group and 24 (20%) in the SC group (RR 0.7; 95% [CI] 0.39 to 1.2; *p* = 0.2). No significant difference was found between the two groups concerning natriuretic peptides and renal function (*p* > 0.05 for all). The cumulative incidence rate of death from any cause at six months from discharge was 8.7% in the JetEcho group and 11.6% in the SC group (HR 0.63, 95% [CI] 0.3–1.4, *p* = 0.3). In AHF patients, a systematic daily bedside echocardiographic monitoring did not reduce 30-day readmission rate for HF and short-term clinical outcomes.

## 1. Introduction

Heart failure (HF) is a major public health problem and a leading cause of mortality, responsible for an increasingly heavy burden of costs [1]. Acute decompensation of HF (AHF) is the leading cause of hospitalization in people over 65 years old with a 20–25% readmission rate in the first month [2]. The management of AHF remains challenging. In spite of medical progress, hospitalizations constantly increase, mainly because of the complexity of medical evaluation and the frequently numerous comorbidities of patients [3]. Nearly 70% of AHF hospitalizations are associated with congestion [4]. Clinical volume overload assessment is crucial for accurate evaluation and represents a promising therapeutic target. Persistent congestion at discharge has been clearly associated with poor prognosis and increased risk of readmission [5]. One third of HF patients suffer from chronic kidney disease (CKD), and worsening renal function often occurs during HF hospitalization, negatively affecting prognosis [6]. Misinterpretation of renal function might result in inappropriate discontinuation of disease-modifying HF therapies, premature discontinuation of decongestive therapies or inappropriate volume adaptations responsible for renal injury [7]. No systematic methods to assess congestion and subsequent tailored therapy during AHF hospitalization are proposed in the available international guidelines [8]. Natriuretic peptides concentration is insufficient for routine treatment guidance [9]. However, echocardiography has been shown to be superior in estimating left ventricular filling pressures (LVFP) and volume status compared to clinical or laboratory parameters [10,11]. Portable cardiac ultrasound devices, with their substantially lower costs, ease of use, and wide availability offer the opportunity for serial assessment of volume status in patients admitted for AHF [12,13].

To date, this study is the first prospective randomized study evaluating the interest of bedside echocardiography on monitoring AHF therapy and its impact on morbidity and mortality. We hypothesized that a simple, fast, daily bedside echocardiographic monitoring (JetEcho) could provide a better estimate filling pressure and volume overload [14], guiding a more suitable therapeutic adjustment and leading to a better post discharge prognosis, especially the early re-admission rate.

## 2. Materials and Methods

### 2.1. Study Oversight

The JECICA trial is a parallel, prospective, randomized open-label study, evaluating an adjusted treatment strategy guided by individualized echocardiographic HF monitoring in patients hospitalized for AHF in Nîmes and Montpellier university hospitals. The rationale and design of the study have been previously detailed [15]. In brief, patients were randomized 1:1 to two parallel groups: a standard of care (SC) group with usual clinical and biological evaluation according to international guidelines [8] versus a group with a daily bedside echocardiographic (Cx50, Philips healthcare, Andover, MA, USA) monitoring during hospitalization (JetEcho group) in addition to standard of care. Both groups were matched by center, age and Left Ventricular Ejection Fraction (LVEF).

The study was approved by the French institutional review board CPP SUD MEDITERRANEE III on 11 July 2016 and was registered at clinicaltrials.gov (NCT02892227) on 8 September 2016. The study conforms with the principles outlined in the Declaration of Helsinki and all patients enrolled in the study provided written consent.

### 2.2. Study Design

An echocardiography was performed on all patients at admission in accordance with standard of care to measure baseline LVEF and verify exclusion criteria. Eligibility requirements included an age ≥ 18 years, hospitalization for AHF receiving at least 40 mg intravenous furosemide, LVEF less than 50% and NT-ProBNP value ≥ 1200 pg per milliliter. Exclusion criteria included a history of mechanical or biological mitral prosthesis, mitral stenosis, severe valvular heart disease requiring surgery within one-month, chronic renal failure requiring dialysis, high degree atrioventricular block (2:1 and 3rd degree), hypertrophic cardiomyopathy, cardiogenic shock (defined by systolic blood pressure (SBP) < 90 mm Hg for ≥30 min or use of pharmacological and/or mechanical support to maintain an SBP ≥ 90 mm Hg, evidence of end-organ hypoperfusion included urine output of <30 mL/h, cool extremities, altered mental status, and/or serum lactate > 2.0 mmol/L), poor echogenicity preventing assessment of LVEF and/or Doppler measurements.

Standard clinical and laboratory evaluations were performed in both groups and were analyzed at days 0, 7 and 30. Therapeutic adaptation was performed in accordance with the current guidelines [8]. In the intervention group, an additional JetEcho was performed by the attending cardiologist during the patient’s daily examination for the duration of the hospitalization; the procedure was rapid to perform (less than five minutes). This bedside echocardiographic monitoring allowed assessment of the transmitral flow and inferior vena cava diameter and variation to estimate filling pressure and volemia [14]. Treatment, especially diuretics, was adjusted based on these JetEcho parameters and according to an algorithm published elsewhere [15] (Appendix A). If necessary, in case of rapid or severe worsening of clinical status, patients in the SC group could benefit from emergency echocardiography.

Patients were evaluated during medical visit at one month after discharge and were contacted by phone at six months. Biological samples were also collected and stored in each center for subsequent analysis of biomarkers at inclusion and one-month post-discharge in both groups.

### 2.3. Outcomes

The primary outcome was the 30-day (±3 days) readmission rate for HF [15].

Secondary outcomes were all-cause mortality at six months, worsening HF during hospitalization (increased dyspnea or need for oxygenotherapy, increased dosages of use isosorbide dinitrate or increased dosages of diuretics or need for IV use), duration of hospital stay, change in weight, evaluation of myocardial (troponin, Nt-proBNP), renal function (GFR, creatinine, urea), and liver biomarkers (AST, ALT, bilirubin) during hospitalization at days 0, 7 and 30 (respectively, D0, D7, D30).

Patients were randomized by blocks of random size using Inclusio software. Investigators were informed of patient group through identification numbers. All readmissions for heart failure were checked and adjudicated by a monitoring board.

### 2.4. Statistical Analysis

Assuming a 15% decrease in 30-day rehospitalization rate in the JetEcho group, with an alpha risk fixed at 5%, a power fixed at 85% in bilateral situation and a drop-out rate of 10%, the sample size was set at 250 subjects (125 per group).

Patient characteristics are described at baseline and by randomization group (JetEcho and SC groups). We performed statistical analyses reporting median and interquartile range for continuous variables, and frequency and percent for qualitative ones. We estimated the primary study outcome as relative risk (RR) of hospitalization 30 days following hospital discharge for JetEcho group compared to SC group.

For the secondary outcomes, we performed statistical comparison between groups using appropriate tests; Pearson Chi-square or Fisher tests for qualitative variables; Student tests or non-parametric Wilcoxon test for continuous variables. We performed further analyses considering time-to-event for the primary outcome and the overall mortality since discharge. We used Kaplan-Meier method and cumulative incidence function for comparison between groups using log-rank or Wald test. All tests are two-sided, with the analyses performed using SAS (SAS Institute, Cary, NC, USA) version 9.4.

## 3. Results

From 15 November 2016 to 7 December 2018, 250 patients admitted for AHF in the university hospitals of Nîmes and Montpellier were recruited into the study. Of these patients, 128 were randomized in the JetEcho group and 122 in the SC group (Figure 1).

One patient from each group withdrew consent prior to allocation and two patients in the JetEcho group withdrew consent after allocation. One patient of the SC group was lost to follow-up. Three patients in the JetEcho group and one in the SC were excluded from analysis because of hospitalization longer than 30 days, as planned in the protocol. The baseline patient characteristics were similar between the two groups (Table 1) except for atrial fibrillation, more frequent in the SC group.

Patients had a median age of 75 years (65;83). The majority of patients presented with LVEF < 40% (77%) and median Nt-proBNP was similar in the two groups. De novo HF was observed in 42% of the patients (106/248). The therapies for HF were similar between the groups at baseline (Table 1). In SC group, 24% of patients had emergency echocardiography leading to treatment adjustment during hospitalization. In the JetEcho group, all patients (100%) had daily echocardiography-guided treatment adjustment and only 9% of the patients had additional emergency echocardiography. At discharge, the treatment target dose percentages were, respectively, 48.1% for beta-blockers in the JetEcho group vs. 41.5% in SC (*p* = 0.4), 50.2% vs. 43.8% for angiotensin-converting-enzyme (ACE) inhibitors/angiotensin receptor blockers (ARB)/angiotensin receptor neprilysin inhibitor (ARNI) (*p* = 0.6), and 52.4% vs. 48.5% for mineralocorticoid receptor antagonist (MRA) (*p* = 0.7). Mean diuretic dosages at end of hospitalization were numerically higher in the JetEcho group, 144.7 mg/day (±15.1) vs. 116.7 mg/day (±12.5) in the SC group (MD 28.0 ± 19.6, *p* = 0.3) (Appendix B).

### 3.1. Primary Endpoint: Readmission Rate for HF at Day 30

The primary endpoint did not differ between both groups. Twenty-two patients (19%) were readmitted in the JetEcho group and 17 (15%) in the SC group at 30 days (RR 1.26; 95% [CI] 0.70 to 2.24; *p* = 0.4) (Table 2).

### 3.2. Secondary Outcomes

Worsening HF occurred in 17 (14%) patients in the JetEcho group and 24 (20%) in the SC group, without statistically significant difference (RR 0.7; 95% [CI] 0.4 to 1.2; *p* = 0.2). The median length of stay during the index hospitalization was similar in the two groups (6 days, (4;10) *p* = 0.8). There was no weight difference from inclusion to discharge between the two groups (−0.40 kg (95% CI; −1.6–0.82) *p* = 0.5). Mean diuresis was no different with 1723 mL/24 h in the JetEcho group and 1766 mL/24 h in the SC group (*p* = 0.6). Other secondary outcomes of biological changes in values at D7 and D30 for, eGFR, blood urea nitrogen (BUN), Nt-proBNP, and troponin did not differ significantly between the two groups (Figure 2). The doses for diuretics (but also regarding the optimal doses of ACEs or MRA) were numerically higher in the JET echo group at discharge when compared to the standard group but these differences do not reach statistical significance (Appendix B).

### 3.3. Six-Month Mortality

The cumulative incidence of death from any cause at six months from discharge was not different between groups with 8.7% in the JetEcho group and 11.6% in the SC group (HR 0.63, 95% [CI] 0.3–1.4, *p* = 0.3) (Figure 3).

## 4. Discussion

In this prospective randomized trial involving patients hospitalized for AHF with LVEF under 50%, a strategy of treatment adjustment guided by a daily, fast, simple, bedside echocardiography in addition to usual clinical and biological evaluation did not reduce the risk of 30-day readmission for HF.

In-hospital worsening heart failure, length of stay, weight loss, mean diuresis during hospitalization, renal and HF biomarkers were also not statistically different in the two groups as 6-month death from any cause.

Data evaluating therapeutic strategies in acute heart failure remain controversial or limited to specific populations. Positive results have been previously published on echocardiography-based strategy for New York Heart Association (NYHA) class I and II outpatients with chronic HF [16]. In the CHAMPION trial, the addition of pulmonary artery pressure evaluation with a wireless implantable hemodynamic monitoring system was associated with a large reduction in hospitalization for HF patients in NYHA class III, but these were chronic ambulatory HF patients [17]. At contrary, in the setting of AHF, the addition of pulmonary artery catheter evaluation to clinical assessment did not affect mortality or hospitalization in the ESCAPE trial [18]. Only one small pilot study reported encouraging results with a treatment strategy guided by echocardiography and lung ultrasound in AHF patients but remained based on non-randomized design with un-balanced population during sequential time periods [19]. All recent studies in AHF evaluating medical treatment (e.g., RELAX-AHF-2 [20] and TRUE-AHF [21]) or guided-treatment strategies, especially based on Nt-proBNP (e.g., PRIMA II trial [9]) have failed to show clinical benefit. Our study was designed to evaluate if a bedside daily echocardiographic monitoring could improve management of AHF during hospitalization. Although there is a strong correlation between residual congestion and prognosis, daily echocardiography for evaluation of filling pressure and congestion did not demonstrate superiority in reducing hospitalization or improving short-term outcomes. However, the excellent acceptability of echocardiography both by patients and physicians should be highlighted, because of easy and quick realization, and the absence of side-effects.

The primary endpoint was evaluated at 30 days as this timeframe is considered the vulnerable period from a pathophysiological and clinical point-of-view with a poor post-discharge outcome [22,23]. Furthermore, it makes sense from a medico-economic point of view since in the United States for example, hospitals with excess 30-day readmission rates are penalized (Hospital Readmissions Reduction Program) [24].

In the JECICA trial, the hospitalization duration is relatively short compared to the national median duration [2]. This short duration could have prevented optimized management, and hence a diminished impact of our strategy. Although there could be difficult in the current medico-economic context, a longer hospital stays [25], could allow a better treatment adjustment and optimal decongestion.

Furthermore, the trial was performed in two large university centers with dedicated HF units with experience of serious HF patient management. The contribution of echocardiography could subsequently be less beneficial than expected considering the expertise of the centers involved in the management of these patients. In addition, several actions to reduce hospitalization for HF had already been implemented in the centers, including telemonitoring, nurse follow-up with a specific HF e-learning training and a coordination book, specific HF medical visits with medicine titration and pharmacists, dietician interventions and early follow-up. All these measures could have changed patients’ management, reduced readmissions and subsequently reduced the potential benefit of a strategy of treatment adjustment guided by echocardiographic monitoring.

Moreover, the potential impact of the JetEcho strategy could be minimized by the severity of the population of the study. First, median NT-proBNP levels were very high in our population at admission, with a value superior to 6.000 pg/mL. In the COPERNICUS study and ADHERE registry, NT-proBNP or BNP were consistently associated with increased risk of mortality and hospitalization for HF, even in patients without congestive symptoms [26,27]. Secondly, the proportion of patients with right HF (global and/or isolated right HF) was more than 60% in our study. Indeed, management of right HF, frequently associated with cardiorenal syndrome and refractory congestion despite diuretic therapy [28], remains complex and challenging.

Although it was not prespecified in the study statistical plan [15], additional analyses evaluating readmissions for cardiac or renal causes at six months showed a significant lower rate of rehospitalizations in the JetEcho group. These promising results could be partially explained by the numerically higher doses of diuretics, ACEs and MRA on the JetEcho group even if these findings remain exploratory.

## 5. Limitations

We should acknowledge some limitations. For practical and ethical reasons, the trial could not be blinded. An urgent echocardiography was performed in a significant proportion of patients (24%) of the standard group. It may have contributed to minimize the expected difference on clinical events between the two studied groups despite a randomized design. The population is relatively small and only included patients from two large university centers with dedicated HF patient pathway. In this trial no patient was implanted with Cardiomems^®^ system (wireless implantable hemodynamic monitoring system for pulmonary artery pressure evaluation) because the device is not refunded in France. Telemonitoring HF program (daily body weight measurement, daily recording of HF symptoms, and personalized education) started in France at the very end of the study and no patient of the study has been included in this telemonitoring program. However, the telemonitoring programs were on the edge to start at the very end of the recruitment and local organization had been improved consistently.

In smaller centers HF patients are not systematically managed by HF specialists, but also by general cardiologists and other specialists, leading to heterogeneous practices and different prognosis [29].

The readmission rate reported in our study is lower than the literature and the effect of the JetEcho strategy on the readmission rate was probably too optimistic. This could have underestimated the sample size calculation and unpowered our study. Indeed, to date, no similar prospective randomized study has been performed and this estimation was subsequently based on previous registries [30]. The daily dosages of therapies, could not be exhaustively recorded because of the complexity of the treatment changes, sometimes occurring several times a day. Finally, the evaluation of LVFP remains challenging and limitations should be acknowledged in some specific populations as patients with AF or hypertrophic cardiomyopathy. However, the JECICA trial aimed to evaluate a bedside and quick monitoring, easy to use for all cardiologists, without required specific imaging expertise. The strategy evaluated was subsequently based on the more reproducible echocardiographic parameters using in routine clinical practice as E/A ratio to evaluate LVFP and IVC size and compliance to assess volume overload and adapt diuretic treatment. Pulmonary congestion evaluated by lung ultrasound has become widely used in recent years. This technique represents a useful method of tracking dynamic changes in pulmonary congestion (B-lines) in response to treatment, and residual congestion at time of discharge could identify patients at high risk for recurrent HF events [31]. A prospective randomized study, based on a more complete approach, combining several rapid parameters, including especially lungs (B-lines), pulmonary artery systolic pressure, kidneys (intrarenal venous flow), and venous system (internal jugular vein diameter) could be interesting for future HF trials [32].

## 6. Conclusions

In this prospective randomized trial involving patients hospitalized for AHF with LVEF under 50%, a strategy of treatment adjustment guided by a fast, simple, bedside, daily echocardiographic monitoring added to the usual clinical and biological evaluation did not reduce the risk of 30-day readmission rate for HF, neither worsening heart failure, renal and HF biological markers as mortality rate.

## Figures and Tables

**Figure 1 jcm-11-02047-f001:**
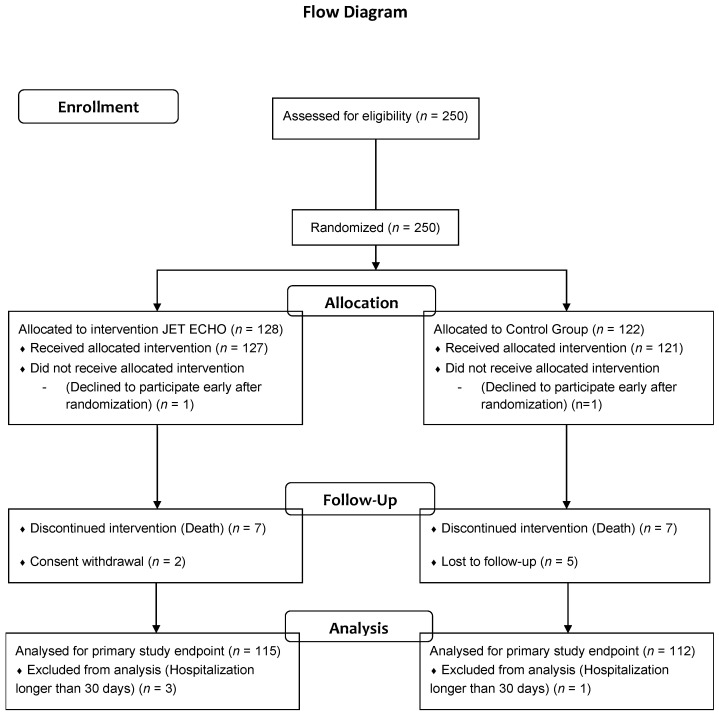
Study Flow Chart.

**Figure 2 jcm-11-02047-f002:**
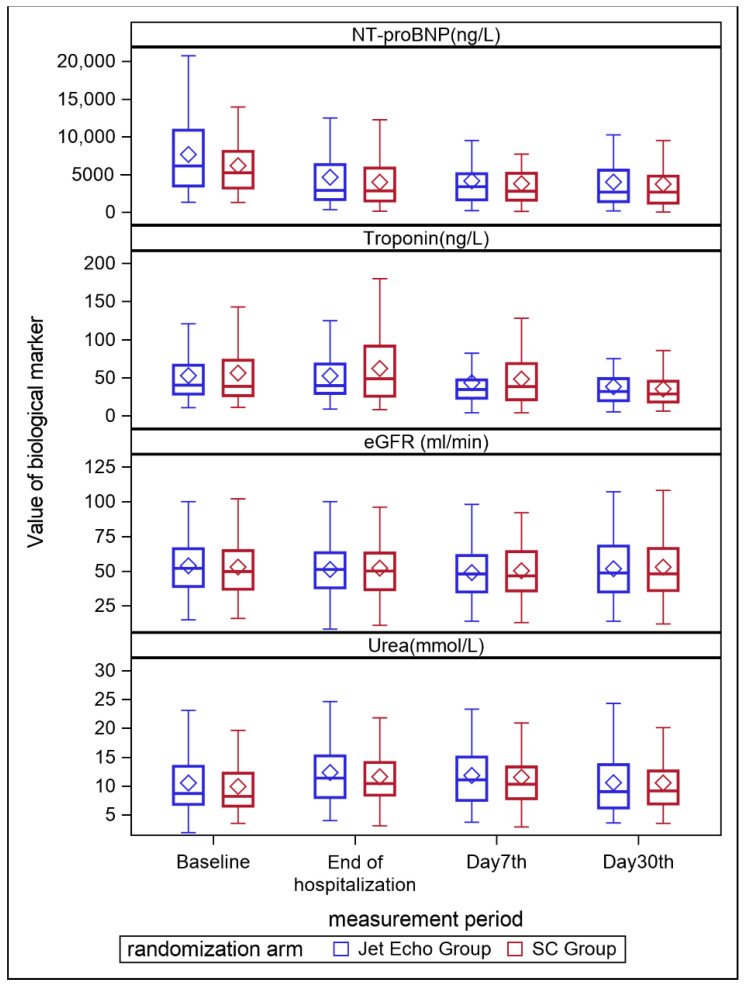
Evolution of biological markers at baseline, end of hospitalization, D7 and D30 after discharge eGFR, estimated glomerular filtration rate; Nt-proBNP, N-terminal pro-B-type natriuretic peptide.

**Figure 3 jcm-11-02047-f003:**
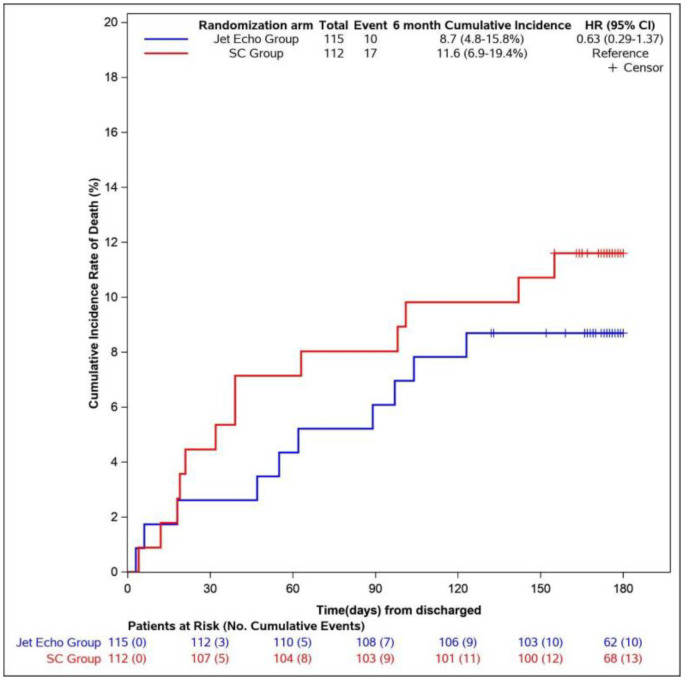
Cumulative Incidence of death from any causes at 6 months from discharge. HR = Hazard ratio. CI = Confidence interval.

**Table 1 jcm-11-02047-t001:** Patient characteristics and HF treatments at baseline.

N(%) or Median (q1;q3)	Jet Echo *N* = 127	Standard Care *N* = 121	*p*-Value *
**Baseline characteristics**			
Age at inclusion (y)	75 (66;82)	76 (64;84)	0.57
Female	38 (30%)	36 (30%)	0.98
BMI (q1;q3)	27.05 (24.2;30.7)	27.35 (24.1;31.1)	0.66
Hypertension	83 (65%)	85 (70%)	0.41
Diabetes	54 (43%)	43 (36%)	0.26
Smoker	17 (13%)	23 (19%)	0.23
Atrial fibrillation	45 (35%)	53 (44%)	0.32
Ischemic Cardiomyopathy	53 (42%)	52 (43%)	1.00
**Clinical parameters**			
NYHA functional class			0.90
stage (I, II)	7 (5%)	5(4%)	
stage III	33 (26%)	33 (28%)	
stage IV	87 (69%)	82 (68%)	
Sinus Rhythm	85 (67%)	61 (50%)	0.01
Heart rate (bpm)	87 (72;106)	82 (72;102)	0.22
SBP (mmHg)	126 (112;141)	125 (110;139)	0.52
DBP (mmHg)	75 (67;89)	76 (67;86)	0.52
LVEF (%)			0.81
<40%	97(76%)	94 (78%)	
40–49%	30 (24%)	27 (22%)	
Types of HF			0.53
Global	68 (54%)	71 (59%)	
Right	6 (5%)	3 (2%)	
Left	53 (42%)	47 (39%)	
**Laboratory**			
Nt-proBNP (ng/L)	6460 (3551;12,336)	6099 (3335.5;12,457)	0.55
Troponin (ng/L)	51.7 (32;184.1)	43.9 (27.7;101.8)	0.22
eGFR (ml/min)	52 (39;66.1)	50 (37;66)	0.76
BUN (mmol/L)	9 (6.8;13.8)	8.3 (6.5;13.2)	0.55
Natremia (mM)	140 (137;143)	141 (138;143)	0.42
Kaliemia (mM)	4.2 (3.8;4.6)	4.1 (3.8;4.5)	0.35
Haemoglobin (g/dL)	12.6 (11.3;14.1)	13 (11.5;14.7)	0.16
**Treatment on admission**			
Diuretic	23 (18%)	23 (19%)	0.86
Beta-blocker	70 (55%)	63 (52%)	0.63
ACE inhibitor	50 (39%)	39 (32%)	0.24
ARB	21 (17%)	19 (16%)	0.86
MRA	21 (17%)	12 (10%)	0.13
ICD	16 (13%)	11 (9%)	0.41
CRT	9 (7%)	4 (3%)	0.25

* *p*-value: Chi-square or Fisher for qualitative variable; Student or Wilcoxon for continuous. Abbreviations: ACE, angiotensin-converting enzyme; ARB, angiotensin II receptor blocker; BMI, body mass index; BUN, blood urea nitrogen; CRT, cardiac resynchronization therapy; DBP, diastolic blood pressure; eGFR, glomerular filtration rate; HF, heart failure; ICD, intracardiac defibrillator; LVEF, left ventricular ejection fraction; RA, mineralocorticoid antagonist; NT-pro-BNP: N-terminal-pro-B-type natriuretic peptide; NYHA, New York Heart Association; SBP, systolic blood pressure; SC, standard of care.

**Table 2 jcm-11-02047-t002:** Primary and secondary outcomes.

Outcomes	Jet Echo *N* = 115	Standard Care *N* = 112	Relative Risk or Hazard Ratio or Difference (95% CI)	*p*-Value
**Primary outcome**				
30-day readmission rate for HF	22 (19%)	17 (15%)	1.26 (0.70–2.24)	0.4
**Secondary outcomes**				
Cumulative incidence of death from any causes at 6 months	8.7%	11,6%	0.63 (0.3–1.4)	0.3
Worsening HF	17 (14%)	24 (20%)	0.7 (0.4–1.2)	0.2
Length of stay (days) *	6 (4;10)	6 (4;10)	0 (−1–1)	0.8
Weight difference from inclusion to discharge (kg) **	−3.22 (±5.1)	−2.83 (±4.26)	−0.40 (−1.6–0.82)	0.5

* Length of stay expressed as median and interquartile range. ** Weight difference as mean and standard deviation (±SD).

## Data Availability

Not applicable.

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
