# Peer review of "Impact of Daily Bedside Echocardiographic Assessment on Readmissions in Acute Heart Failure: A Randomized Clinical Trial"

_jcm, 2022, doi:10.3390/jcm11072047_

Round 1

Reviewer 1 Report

In this prospective RCT involving AHF patients with LVEF less than 50%, a strategy of treatment adjustment guided by a daily, fast, simple, bed-side TTE in addition to usual clinical and biological evaluation did not reduce the risk of 30-day readmission for HF.

Although this is a negative study, the study is still important to consider the strategy for AHF.

There are some drawbacks that have caught may attention.

  1. Please discuss the rationale of the therapeutic algorithm in this study. The evaluation of LVFP by E/A in sinus rhythm seems to be too simple.
  2. Please show the number of the patients admitted for acute heart failure in these institutes during the enrollment period (e.g., the number of patients who were excluded in each exclusion criteria). These data is important to consider the generality of the results of this study.
  3. Please show the frequency of emergency echocardiography in case of rapid or severe worsening of clinical status in the SC group.
  4. The details of the definition of HF rehospitalization need to be described. Is there any adjudication system for HF readmission (primary outcome)?
  5. Why did the authors focus only on the patients with LVEF less than 50%? HFpEF patients also could be the candidate for this project.

Reviewer 2 Report

In Table 1 P values for the two-group comparison should be reported.

"to date, no treatment strategies have been demonstrated to be beneficial for patients hospitalized for AHF" what do the Authors mean? that ultrasound-guided treatment strategies have not been demonstrated to be beneficial in this clinical setting? or any treatment strategy has not been demonstrated to be beneficial?  Please, try to be not misleading.

"Although there is a strong correlation between residual congestion and prognosis, daily echocardiography for evaluation of filling pressure and congestion did not demonstrate superiority in reducing hospitalization or improving short-term outcomes" references are needed.

"To date, this study is the first prospective randomized study evaluating the interest 61 of bedside echocardiography on monitoring AHF therapy and its impact on morbidity and mortality." Please discuss other similar studies with different design (e.g., ESC Heart Fail. 2018 Feb; 5(1): 120–128.)

Reviewer 3 Report

The manuscript by Ricci et al. entitled “Impact of daily bedside echocardiographic assessment on readmissions in acute heart failure: a randomized clinical trial” aimed to o assess whether daily bedside echocardiographic monitoring (JetEcho) improved outcomes in AHF. The article is well written and leads some evidence to such point; however, some major issues need to be addressed to improve the significance and reliability of the results of the study:

-Firstly, why was evaluated only transmitral flow and not trantricuspid flow and PASP?

-In Table 1 p-value should be reported for all variables. Moreover, important baseline data as DBP or PASP are not reported. Please, add this data.

- In “Treatment on admission” section of Table 1, ARNI are not reported. Were there any patients taking ARNI?

- In the past years, different devices have been investigated to help in identifying early decompensation events in patients with heart failure and reduced ejection fraction, reducing hospital admissions. Were there any patients with remote haemodynamic monitoring device? Please, clarify in the text, because this is an essential parameter.

-An important parameter was not consider: was non-invasive ventilation (NIV) an exclusion criteria? If no, please specify how many patients in each group required NIV? There were any differences between the two groups?

Minor iusses:

-In line 88 there is a typo. Please, check this part.

Round 2

Reviewer 1 Report

Thank you very much for appreciating the comments and act on them.

Author Response

On behalf of my co-authors, we would like to thank you for your thorough review and helpful comments on our paper and for giving us the opportunity to resubmit a revised manuscript. 

Reviewer 3 Report

The value of PAPS is critical in assessing a patient's echocardiographic congestion.
In addition, this parameter is not difficult to estimate, so it is advisable to add it
to the algorithm for patient assessment.

Author Response

We agree with the reviewer that high PAPs are associated with clinical congestion and poor prognosis. However, as we mentioned in our previous response, we do not have these data in our study as we choose only two simple parameters (mitral flow and IVC) to propose a fast, reliable, acceptable, with wide scientific publications, to propose an algorithm accessible to most part of physicians, experimented, as well as junior doctors, heart failure specialists as well as other cardiologists, with the hypothesis that if the study was positive, it could be propose to other specialists dealing with heart failure. Besides, tricuspid regurgitation is not always easily obtained in a daily, fast, bedside configuration. At least, inaccuracy of Doppler echocardiographic estimation of PAPs, compared with right heart catheterization, has been reported several times and should probably not be relied on to make the diagnosis of pulmonary hypertension or to follow the efficacy of therapy (Inaccuracy of Doppler echocardiographic estimates of pulmonary artery pressures in patients with pulmonary hypertension: implications for clinical practice, Jonathan D Rich et al, Chest. 2011 May;139(5):988-993. doi: 10.1378/chest.10-1269. Epub 2010 Sep 23). As we mentioned in the discussion section, studies evaluating a multiparameter analyse of congestion could be proposed in the future in the setting of AHF, with the difficult balance of finding a real life, applicable strategy. We propose to add this sentence in the limits section (page 9, lines 332-336): “A prospective randomized study, based on a more complete approach, combining several rapid parameters, including especially lungs (B-lines), pulmonary artery systolic pressure, kidneys (intrarenal venous flow), and venous system (internal jugular vein diameter) could be interesting for future HF trials [32] ».